# Profiling of inflammatory cytokines in patients with caustic gastrointestinal tract injury

Hao-Tsai Cheng[1,2,3,4], Chen-June Seak[ID][3,5], Chien-Cheng Cheng[3,4,6], Tsung-Hsing Chen[2,3,4], Chang-Mu Sung[2,3,4], Shih-Ching Kang[7], Yu-Jhou Chen[ID][2,3], Chip-Jin Ng[3,5], Chao-Wei Lee[8], Shu-Wei Huang[1,3], Hsin-Chih Huang[2,3], Tzung-Hai Yen[3,9]*

1 Division of Gastroenterology and Hepatology, Department of Internal Medicine, New Taipei Municipal TuCheng Hospital, New Taipei City, Taiwan, 2 Department of Gastroenterology and Hepatology, Chang Gung Memorial Hospital, Linkou, Taiwan, 3 College of Medicine, Chang Gung University, Taoyuan, Taiwan, 4 Graduate Institute of Clinical Medicine, College of Medicine, Chang Gung University, Taoyuan, Taiwan, 5 Department of Emergency Medicine, Chang Gung Memorial Hospital, Linkou, Taiwan, 6 Department of Medical Image and Intervention, Chang Gung Memorial Hospital, Linkou, Taiwan, 7 Division of Trauma and Emergent Surgery, Chang Gung Memorial Hospital, Linkou, Taiwan, 8 Division of General Surgery, Department of Surgery, Chang Gung Memorial Hospital, Linkou, Taiwan, 9 Department of Nephrology, Clinical Poison Center, Kidney Research Center, Center for Tissue Engineering, Chang Gung Memorial Hospital, Linkou, Taiwan

* m19570@adm.cgmh.org.tw

**Data Availability Statement:** All relevant data are within the manuscript.

## Abstract

### Introduction

Study of inflammatory cytokines in patients with caustic gastrointestinal tract injury is sketchy. This study investigated the cytokine profiling of patients with caustic substance ingestion, and analyzed the differences between patients with severe and mild injury.

### Methods

This prospective, cross-sectional study enrolled 22 patients admitted to Chang Gung Memorial Hospital between March and October 2018. All patients underwent esophagogastroduodenoscopy in 24 hours. Patients were categorized into two subgroups, as mild (<2b, n = 11) or severe ($\geq$2b, n = 11) group.

### Results

The neutrophil count was higher in severe than mild group (P = 0.032). Patients in mild and severe groups exhibited significantly higher circulating inflammatory cytokines than healthy control, including interleukin (IL)-2, IL-5, IL-8, IL-9, IL-12, IL-13, interferon-gamma inducible protein-10, macrophage inflammatory protein-1 beta, regulated upon activation, normal T cell expressed and presumably secreted and tumor necrosis factor-alpha. Furthermore, the levels of IL-2 and tumor necrosis factor-alpha were significantly higher in patients with severe group than mild group. Although there was no difference in cumulative survival between both groups (P = 0.147), the severe group received more operations (P = 0.035) and suffered more gastrointestinal complications (P = 0.035) than mild group.

**Funding:** This study was funded by research grants from Chang Gung Memorial Hospital (CMRPG3F1931 to Dr. Hao-Tsai Cheng; CMRPG3G0871, CMRPG3G0872, CORPG3K0192 to Dr. Tzung-Hai Yen). The funding sources had no role in the study's design, conduct, or reporting.

**Competing interests:** The authors have declared that no competing interests exist.

## Conclusion

Caustic substance ingestion produces mucosal damages and leads to excessive neutrophils and inflammatory cytokines in peripheral blood.

## Introduction

Caustic substance ingestion is an uncommon but life-threatening condition worldwide [1]. It causes a wide spectrum of damages on aero-digestive tract and in turn various complications that are challenging to manage [2]. The severity of damages on gastrointestinal tract is associated with the caustic amount, property, concentration, and type of ingested substances [3–5]. According to the 2019 annual report of the American Association of Poison Control Centers, there were 185139 cases of exposure to cleaning substance [6]. The total number of caustic injury cases in Taiwan between 1996 and 2010 according to National Health Insurance Research Database was 16,001 (8,991 female patients, 7,010 male patients) [7]. In Iran, up to 33% (115 patients) cases requiring surgery [8].

Computerized tomography scan of the chest and abdomen has been reported as alternative diagnostic modality to esophagogastroduodenoscopy for the estimation of caustic injury [9,10]. Zagar's classification is widely used in caustic patient for staging the damage of gastrointestinal tract. Over two-thirds of patients with severe injury ($\geq$2b) in gastrointestinal tract have been shown to have higher comorbidity and mortality [11,12]. Most of them required active treatments including dilatation or surgery. Severe caustic damage of gastrointestinal would often reduce the patient's quality of life. Corticosteroids have been prescribed for caustic injury patients to prevent the stricture formation of gastrointestinal tract; however, the benefits of corticosteroids remain still controversial [13–18].

Theoretically, caustic injury of gastrointestinal tract mucosa is thought to be associated with a high degree of systemic inflammation or cytokine storm. Nevertheless, study of inflammatory cytokine profile in these patients is still sketchy. Therefore, this study aimed to investigate the cytokine profiling of patients with caustic substance ingestion, and analyze the differences between patients with severe and mild caustic gastrointestinal tract injury.

## Results

A total of 22 patients were enrolled and divided into two subgroups of 11 patients according the mucosal severity of Zagar's classification as mild group ($<$ 2b) or severe group ($\geq$ 2b) (Table 1). Severe group patients presented six patients with grade 3b damage in stomach and/or esophagus. Duodenum was less injured in both groups, although a P value of 0.026 was noted between two groups. There was no different between both groups in endotracheal tube with mechanical ventilation used (P = 1.000). Two patients in mild group still needed ventilator support or prevention for endoscopy.

Basic characteristics of patients were listed in Table 2. The ingested amount of caustic substance was greater in severe group and P value was 0.019 [severe group: 188 ± 130 (20–400), mild group: 58 ± 64 (5–150)]. Severe group patients had severer psychiatric problem (severe group: 10 (90.9%), mild group: 4 (36.4%), P = 0.024]. There were no significant differences in gender, acid/alkaloid, and systemic comorbidities.

The renal function, liver enzyme, C-reactive protein and white blood cell count were described in Table 3. There were no significant differences between both groups including

**Table 1. Esophagogastroduodenoscopy findings of patients with caustic gastrointestinal injury (n = 22).**

| Variable | Severe group (n = 11) | Mild group (n = 11) | P value |
|---|---|---|---|
| Esophagus | | | <0.001* |
| Grade 0, n | 1 | 6 | |
| Grade 1, n | 0 | 5 | |
| Grade 2a, n | 3 | 0 | |
| Grade 2b, n | 0 | 0 | |
| Grade 3a, n | 3 | 0 | |
| Grade 3b, n | 4 | 0 | |
| Stomach | | | <0.001* |
| Grade 0, n | 0 | 8 | |
| Grade 1, n | 0 | 1 | |
| Grade 2a, n | 0 | 2 | |
| Grade 2b, n | 1 | 0 | |
| Grade 3a, n | 6 | 0 | |
| Grade 3b, n | 5 | 0 | |
| Duodenum † | | | 0.026* |
| Grade 0, n | 5 | 11 | |
| Grade 1, n | 2 | 0 | |
| Grade 2a, n | 0 | 0 | |
| Grade 2b, n | 1 | 0 | |
| Grade 3a, n | 1 | 0 | |
| Grade 3b, n | 0 | 0 | |
| The most severe grade | | | <0.001* |
| Grade 0, n | 0 | 6 | |
| Grade 1, n | 0 | 3 | |
| Grade 2a, n | 0 | 2 | |
| Grade 2b, n | 1 | 0 | |
| Grade 3a, n | 4 | 0 | |
| Grade 3b, n | 6 | 0 | |
| Endotracheal tube with mechanical ventilation during esophagogastroduodenoscopy, n (%) | 2 (18.2) | 2 (18.2) | 1.000 |

Note:

† The duodenal mucosa was invisible in two cases of the severe group due to patients' intolerance for examination.

white blood cell count, hemoglobin and platelet. The difference of white blood cell was further analyzed. It was noted that the percentage of neutrophils was higher in severe group (84.2 ± 10.0 versus 69.8 ± 16.6%, P = 0.032) than mild group.

Log-rank test was performed to explore the overall survival outcome and there was no significant difference between two subgroups (P = 0.147, Table 4). Two patients expired in severe group due to acute myocardial infarction and respiratory failure combined sepsis. Compared with mild group, the severe group received more operations (P = 0.035) and had more gastrointestinal complication including stricture (P = 0.035). The average length of in-hospital stay was significantly longer in severe group compared with that in mild group (24.4 ± 20.4 versus 6.6 ± 9.0 days, P = 0.003). There were no significant differences between both group in systemic complications, ICU admission and medications used.

Clinical courses of 5 patients with caustic gastrointestinal tract injury who underwent surgery were presented in Table 5. The indications for surgery were mainly esophageal stricture or gastric outlet obstruction.

**Table 2. Baseline demographics of patients with caustic gastrointestinal tract injury (n = 22).**

| Variable | Severe group (n = 11) | Mild group (n = 11) | P value |
|---|---|---|---|
| Age, year | 52.3 ± 18.1 (18–80) | 48.6 ± 24.3 (25–93) | 0.356 |
| Female, n (%) | 6 (54.5) | 4 (36.4) | 0.392 |
| Caustic substances | | | |
| Property | | | 0.395 |
| Acid, n (%) | 4 (36.4) | 6 (54.5) | |
| Alkaline, n (%) | 7 (63.6) | 4 (36.4) | |
| Neutral, n (%) | 0 (0) | 1 (9.1) | |
| Strong caustics (pH < 2 or > 12) | 6 (54.5) | 6 (54.5) | 1.000 |
| Amount, mL | 188 ± 130 (20–400) | 58 ± 64 (5–150) | 0.019* |
| Intentional ingestion, n (%) | 10 (90.9) | 8 (72.7) | 0.586 |
| Previous suicide attempts, n (%) | 2 (18.2) | 1 (9.1) | 1.000 |
| Psychiatric comorbidities, n (%) | 10 (90.9) | 4 (36.4) | 0.024* |
| Depressive disorders, n (%) | 5 (45.5) | 1 (9.1) | 0.149 |
| Adjustment disorder, n (%) | 3 (27.3) | 3 (27.3) | 1.000 |
| Bipolar disorders, n (%) | 2 (18.2) | 0 (0) | 0.476 |
| Schizophrenia, n (%) | 1 (9.1) | 0 (0) | 1.000 |
| Alcohol use disorder, n (%) | 3 (27.3) | 4 (36.4) | 1.000 |
| Systemic comorbidities | | | |
| Hypertension, n (%) | 4 (36.4) | 2 (18.2) | 0.635 |
| Diabetes mellitus, n (%) | 2 (18.2) | 2 (18.2) | 1.000 |
| Fever, n (%) | 1 (9.1) | 2 (18.2) | 1.000 |

Note: Data of continuous variables were expressed as mean ± standard deviation (range), and those of categorical variables were presented as numbers with percentages.

As shown in Figs 1 and 2, patients in mild and severe groups exhibited significantly higher circulating levels of inflammatory cytokines compared with those of healthy control, including interleukin (IL)-2, IL-5, IL-8, IL-9, IL-12, IL-13, interferon-gamma inducible protein-10, macrophage inflammatory protein-1 beta, regulated upon activation, normal T cell expressed and presumably secreted and tumor necrosis factor-alpha. Furthermore, the circulating levels of

**Table 3. Laboratory data of patients with caustic gastrointestinal tract injury (n = 22).**

| Variables | Severe group (n = 11) | | Mild group (n = 11) | | P value |
|---|---|---|---|---|---|
| | Mean ± standard deviation | Range | Mean ± standard deviation | Range | |
| Alanine aminotransferase, U/L | 23.6 ± 7.4 | 15–36 | 31 ± 16.3 | 12–67 | 0.411 |
| Creatinine, mg/dL | 0.79 ± 0.19 | 0.52–1.15 | 1.06 ± 0.68 | 0.58–3.05 | 0.199 |
| Estimated glomerular filtration rate, mL/min/1.73m$^2$ | 102.7 ± 38.0 | 55.9–174.3 | 86.1 ± 30.2 | 15.1–134.3 | 0.562 |
| C-reactive protein, mg/dL | 41.0 ± 27.2 | 1.0–67.7 | 42.3 ± 43.8 | 1.6–96.5 | 0.699 |
| Hemoglobin, g/dL | 15.3 ± 1.7 | 12.6–18.6 | 14.2 ± 1.3 | 11.9–16.0 | 0.120 |
| Hematocrit, % | 45.7 ± 5.0 | 37.7–54.4 | 42.9 ± 3.1 | 37.9–46.7 | 0.091 |
| Platelet, $10^3$/μL | 245.6 ± 58.2 | 166–367 | 282.6 ± 65.0 | 138–372 | 0.116 |
| White blood cell, $10^3$/μL | 15.4 ± 5.0 | 7.7–25.7 | 11.5 ± 3.4 | 7.5–17.7 | 0.078 |
| Neutrophil, % | 84.2 ± 10.0 | 60.8–96.0 | 69.8 ± 16.6 | 48.0–88.5 | 0.032* |
| Lymphocyte, % | 11.0 ± 9.1 | 2.0–32.0 | 23.0 ± 12.9 | 8.2–42.0 | 0.013* |
| Monocyte, % | 4.0 ± 2.0 | 1.9–8.5 | 4.7 ± 2.4 | 1.0–8.2 | 0.508 |
| Eosinophil, % | 0.5 ± 0.4 | 0.0–1.4 | 2.1 ± 2.2 | 0.0–6.2 | 0.056 |
| Basophil, % | 0.4 ± 0.4 | 0.0–1.0 | 0.5 ± 0.3 | 0.1–1.0 | 0.503 |

**Table 4. Clinical outcomes of patients with caustic gastrointestinal tract injury (n = 22).**

| Variables | Severe group (n = 11) | Mild group (n = 11) | P value |
|---|---|---|---|
| Admission, n (%) | 10 (90.9) | 5 (45.5) | 0.063 |
| Hospitality, day | 24.4 ± 20.4 (3–65) | 6.6 ± 9.0 (1–30) | 0.003* |
| Intensive care unit admittance, n (%) | 3 (27.3) | 1 (9.1) | 0.586 |
| Intensive care unit period, day | 7.3 (3–15) | 8 (8–8) | 1.000 |
| Medication | | | |
| Proton pump inhibitor, n (%) | 11 (100) | 10 (90.9) | 1.000 |
| Histamine 2 blocker, n (%) | 0 (0) | 1 (9.1) | 1.000 |
| Antibiotic, n (%) | 8 (72.7) | 4 (36.4) | 0.087 |
| Operation, n (%) | 5 (45.5) | 0 (0) | 0.035* |
| Hemodialysis for acute kidney injury, n (%) | 0 (0) | 1 (9.1) | 1.000 |
| Systemic complications | | | |
| Aspiration pneumonia, n (%) | 3 (27.3) | 2 (18.2) | 1.000 |
| Respiratory failure, n (%) | 1 (9.1) | 2 (18.2) | 1.000 |
| Hepatic, n (%) | 2 (18.2) | 0 (0) | 0.476 |
| Renal, n (%) | 1 (9.1) | 1 (9.1) | 1.000 |
| Disseminated intravascular coagulation, n (%) | 0 (0) | 0 (0) | 1.000 |
| Gastrointestinal complications, n (%) | 5 (45.5) | 0 (0) | 0.035* |
| Stricture, n (%) | 4 (36.4) | 0 (0) | 0.090 |
| Perforation, n (%) | 0 (0) | 0 (0) | 1.000 |
| Fistula, n (%) | 0 (0) | 0 (0) | 1.000 |
| Bleeding, n (%) | 4 (36.4) | 0 (0) | 0.090 |
| Endoscopic dilation, n (%) | 2 (18.2) | 0 (0) | 0.476 |
| Overall survival | | | 0.147 † |
| 3-month, n (%) | 11 (100.0) | 11 (100.0) | |
| 6-month, n (%) | 9 (81.8) | 11 (100.0) | |
| 12-month, n (%) | 9 (81.8) | 11 (100.0) | |
| Follow-up period, month | 10.3 ± 4.4 (3–16) | 11.9 ± 3.1 (8–16) | 0.537 |

Note: Data of continuous variables were expressed as mean ± standard deviation (range), and those of categorical variables were presented as numbers with percentages.

† The p value of overall survival outcome was obtained by log-rank test.

IL-2 and tumor necrosis factor-alpha were higher in patients with severe group than mild group.

## Discussion

The analysis showed increased circulating levels of inflammatory cytokines (IL-2, IL-5, IL-8, IL-9, IL-12, IL-13, interferon-gamma inducible protein-10, macrophage inflammatory

**Table 5. Clinical courses of 5 patients with caustic gastrointestinal tract injury who underwent surgery.**

| Case | Time between caustic substance ingestion and surgery (day) | Type of surgery | Indication for surgery |
|---|---|---|---|
| 1 | 87 | Gastrojejunostomy and Roux-en-Y | Gastric outlet obstruction |
| 2 | 77 | Feeding jejunostomy | Esophageal inlet stricture |
| 3 | 10 | Feeding jejunostomy and tracheostomy | Esophageal stricture |
| 4 | 15 | Total gastrectomy and feeding jejunostomy | Gastric necrosis with obstruction |
| 5 | 2 | Esophagectomy and total gastrectomy and feeding jejunostomy | Severe corrosive injury |

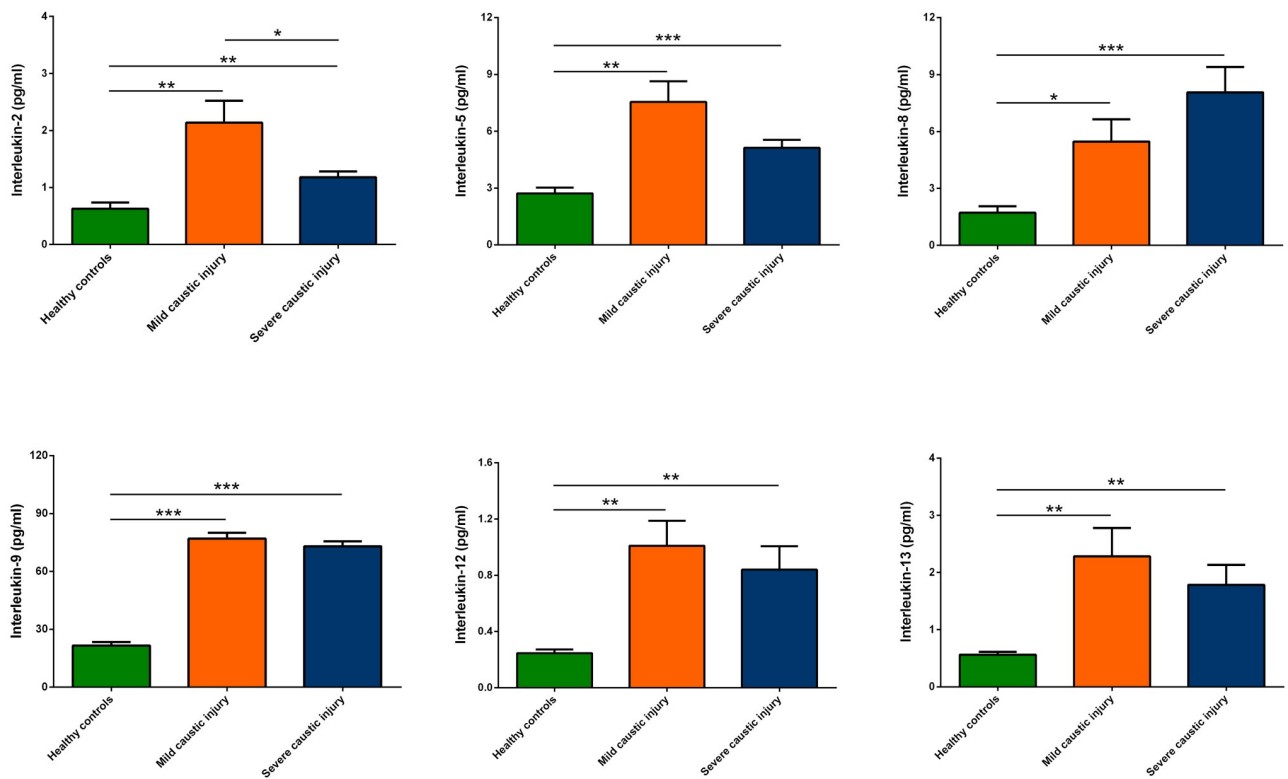

**Fig 1. Multiplex cytokine immunoassay.** Patients in mild and severe groups exhibited significantly higher circulating inflammatory cytokines compared with those of healthy control, including IL-2, IL-5, IL-8, IL-9, IL-12 and IL-13. Furthermore, the circulating IL-2 was higher in patients with severe group than mild group.

protein-1 beta, regulated upon activation, normal T cell expressed and presumably secreted and tumor necrosis factor-alpha) in patients with caustic gastrointestinal injury than healthy controls. In addition, the neutrophil counts and circulating levels of certain cytokines (IL-2 and tumor necrosis factor-alpha) were higher in patients with severe than mild caustic gastro-intestinal injury.

As reported by previous study, the tissue damage by traumatic injuries could produce rapid immune responses. Mediators and cells of adaptive immune systems suffered temporal modification that have been classified to pro-inflammatory and counter-inflammatory and were commonly consulted as systemic inflammatory response syndrome, compensatory anti-inflammatory response syndrome or mixed antagonist response syndrome [19–21]. Elevation of serum IL-6 and IL-10 levels has been reported in the initial phase of the immune response to sepsis. [19] Osuchowski et al [20] showed that plasma concentrations of pro-inflammatory (IL-6, tumor necrosis factor-alpha, IL-1β, kupffer cell, macrophage inflammatory protein-2, monocyte chemoattractant protein-1, and eotaxin) and anti-inflammatory (tumor necrosis factor soluble receptors, IL-10, IL-1 receptor antagonist) biomarkers were increased at the early stage of sepsis in murine model. Nevertheless, no data are available at present for patents with caustic gastrointestinal tract injury. Therefore, it is thought that the changes in circulating cytokine levels might be potential biomarkers for outcome prediction.

The pathophysiology of caustic substance ingestion injuries in gastrointestinal tract depended on a lot of elements, including ingested substance formation, pH, concentration, amount, mucosal surface contact time, viscosity, and presence or absence of food in the

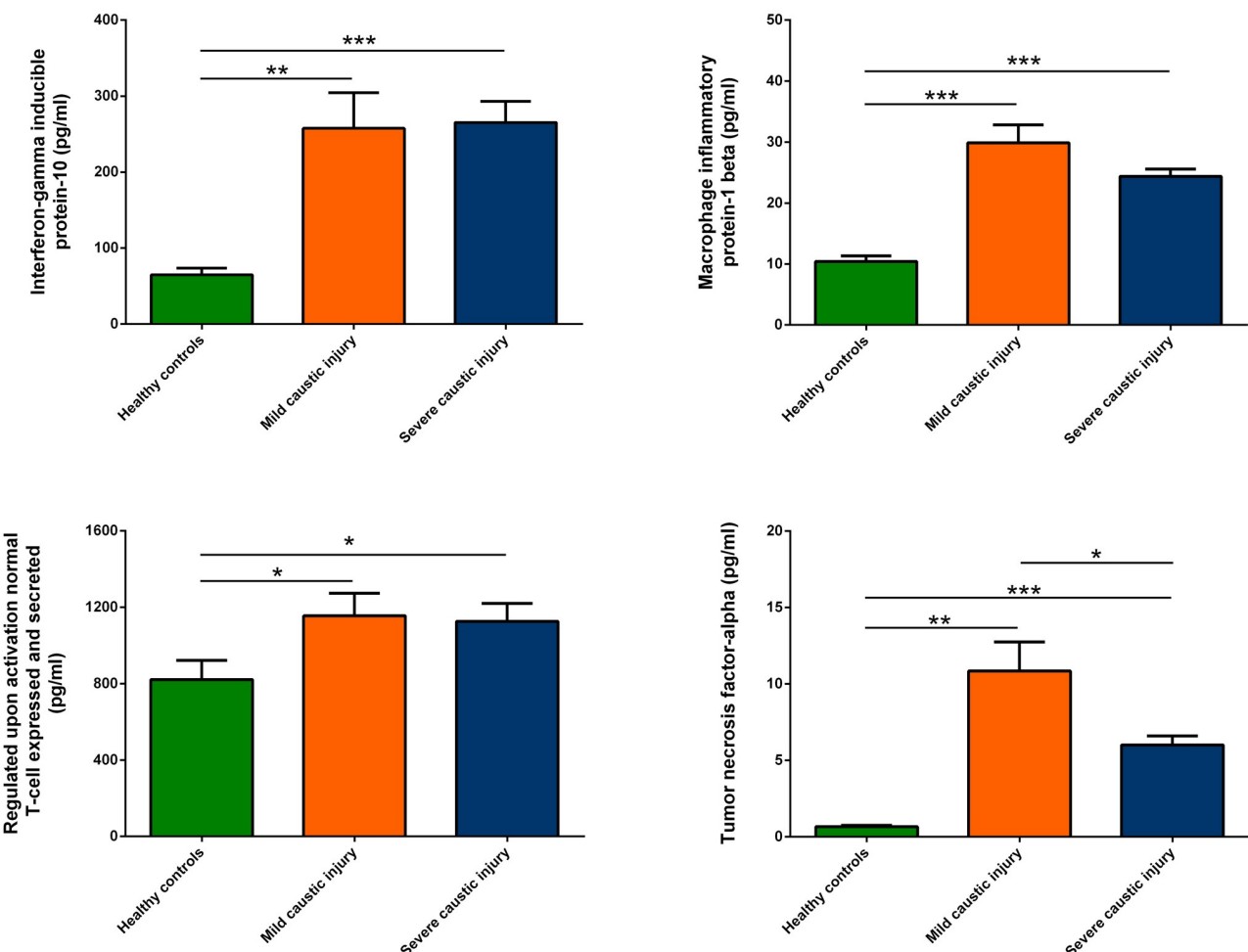

**Fig 2. Multiplex cytokine immunoassay.** Patients in mild and severe groups exhibited significantly higher circulating inflammatory cytokines compared with those of healthy control, including interferon-gamma inducible protein-10, macrophage inflammatory protein-1 beta, regulated upon activation, normal T cell expressed and presumably secreted and tumor necrosis factor-alpha. Furthermore, the circulating tumor necrosis factor-alpha was higher in patients with severe group than mild group.

stomach [1,2,22]. Alkali caustic substance induced saponification or liquefaction mucosa necrosis. Acid caustic substance causes coagulum and penetrates less deeply into exposure gastrointestinal lumen tissue. So, alkali caustic injuries were more severe then acid substance [23,24]. According to our past study, "pH-dependent" in alkalis and "dose-dependent" in acids were the risk of mortality and perforation [25]. High pH of alkalis was a simple result to induce serious injuries of caustic patients, whereas acid substance induced severe gastrointestinal tract damage (Zagar's grade ≥2 b) by both large dose and low pH.

Acid damage tissues by denaturing proteins leading to coagulation necrosis, and alkali-associated damages are caused by saponifying fats of tissues. The coagulation change is to prevent the acid penetrating to deep tissue and limit the damage. Alkalis liquefaction necrosis was easily induced to extending penetration of tissue [1]. According to Nam's study, visceral fat, leptin as well as circulating levels of IL-1 beta and IL-6 were higher in patients with reflux esophagitis than healthy controls. The cytokine changes in caustic gastrointestinal tract damage were different than reflex esophagitis [21].

None of our patients were treated with glucocorticoids. Beneficial effects of steroid on preventing stricture formation in severe caustic ingestion, is controversial [16–18,26,27]. The anti-inflammatory effect of glucocorticoids is suggested to be due to the suppression of nuclear factor kappa-light-chain-enhancer of activated B cells and activator protein 1 activity through interaction with glucocorticoid receptor [28]. IL-10 is considered as a potent immunomodulatory and anti-inflammatory cytokine. In our study, no significant difference in IL-10 levels between two groups of caustic patients. The stricture of caustic injury in the esophagus and stomach was increased by 20.9- and 7.1-fold, respectively, by strong acid and alkalis according our previous study [25]. In our study, 36.4% caustic patients with gastrointestinal stricture only appear in severe group and there was no significant (P = 0.09). It is suggested that use of steroid had no benefit to prevent esophageal stricture after caustic ingestion.

The most obvious limitation in this study was small sample size. Moreover, there was no further analysis for cytokine change in complication such as systemic complication, or gastrointestinal complication such as stricture. Therefore, it was difficult to conclude that there is no benefit of steroid for prevent esophageal stricture owing to no significance difference in IL-10 between two groups of caustic patients. Notably, anti-tumor necrosis factor alpha treatment with infliximab did not ameliorate the degree of fibrosis in alkali burns of the esophagus in the rat [29]. Knowing the cytokine response related complication after caustic injury may evoke more study for treatment. Further studies are necessary to elucidate the influences of immune responses on the clinical outcomes of corrosive patients.

## Conclusion

This is supposed to be the first study to profile inflammatory cytokines in patients with caustic gastrointestinal tract injury. Caustic injury of the upper gastrointestinal tract causes severe mucosal damages and leads to excessive levels of inflammatory cytokines and neutrophils in the peripheral blood. The findings of this translational study provide clinical significance. Our analysis found that caustic gastrointestinal tract injury is characterized by a systemic inflammatory response that involves elevated levels of circulating inflammatory cytokines and neutrophils. The analysis showed increased circulating levels of inflammatory cytokines (IL-2, IL-5, IL-8, IL-9, IL-12, IL-13, interferon-gamma inducible protein-10, macrophage inflammatory protein-1 beta, regulated upon activation, normal T cell expressed and presumably secreted and tumor necrosis factor-alpha) in patients with caustic gastrointestinal injury than healthy controls. Furthermore, the neutrophil counts and circulating levels of certain cytokines (IL-2 and tumor necrosis factor-alpha) were higher in patients with severe than mild caustic gastrointestinal injury. Additionally, since circulating levels of IL-2 and tumor necrosis factor-alpha were higher in nonsurvivors than in survivors, these two cytokines might have clinical potential as promising prognostic markers for caustic gastrointestinal injury. In this context, clinical determination of the circulating inflammatory response, particularly IL-2 and tumor necrosis factor-alpha levels, could serve as a valuable adjunct to physiological predictors for the prediction of poor outcome.

## Materials and methods

### Ethical statement

This study adhered to the Declaration of Helsinki and had been approved by the Medical Ethics Committee of Chang Gung Memorial Hospital. The Institutional Review Board number allocated to the study was 201602045B0. Informed written consent was obtained from all patients according to the guidelines of our institutional review board. Since some of the patients had psychiatric comorbidities, all patients were routinely assessed for capacity to

consent by the principal investigator (H.-T.C.). Patient who had impaired capacity to consent were excluded from this study. The participation in this study was voluntary and patients can opt out at any time. None of the patients underwent treatment at the time of the study. Furthermore, participation in this study did not affect patient access to treatment. The Medical Ethics Committee of Chang Gung Memorial Hospital had approved the study protocol and consent procedure and knowing that some of the patients may have psychiatric comorbidities.

### Patient recruitment—Inclusion and exclusion criteria

Between March 2018 and October 2018, we prospectively enrolled 22 patients, including 12 men and 10 women in Chang Gung Memorial Hospital. As mentioned, patients who refused to sign an informed consent or who had impaired capacity to consent were excluded from this study. All patients underwent esophagogastroduodenoscopy with blood serum collection within 24 hours after admission. The mucosal damage was graded using Zagar's modified endoscopic classification scheme. Patients were divided into two subgroups according the mucosal severity of Zagar's classification: mild group ($< 2b$) and severe group ($\geq 2b$). Laboratory data, including hematology and biochemistry, were collected upon arrival to the emergency department. The ingested caustic compounds were confirmed by referring to the label on containers. Strong caustics were defined as substances with pH $< 2$ or $> 12$. The obtainment of ingested dose and amounts of caustic compounds, intent of ingestion, psychiatric comorbidities, previous suicide attempt records, treatment courses, intensive care unit admittance, and gastrointestinal/systemic complications were recorded for each case.

### Endoscopic survey

Esophagogastroduodenoscopy was available around the clock at Chang Gung Memorial Hospital. The procedure was performed by experienced endoscopists within 24 hours after ingestion. Oral xylocaine spray was used, except in patients that needed ventilation support under general anesthesia for respiratory difficulty or unclear consciousness. Insufflations and retrovision maneuvers were carefully performed or avoided in patients with severe injury. Caustic mucosal damage of the gastrointestinal tract was graded using Zargar's modified endoscopic classification as grade 0, 1, 2a, 2b, 3a or 3b [30].

### Clinical management

Proton pump inhibitors or histamine 2 blockers were prescribed for the caustic injury patients. The patients also received parenteral nutrition without oral intake until their clinical status was regarded as stable. For suspected infection, blood cultures were obtained before the administration of antibiotics. Once a destabilized condition or respiratory difficulty encountered, the patient was transferred to the intensive care unit for critical care. After discharge, patients were followed in the outpatient clinic for at least 6 months.

### Clinical complications

Any observed gastrointestinal or systemic complications were recorded during follow-up. Upper gastrointestinal complications included perforation, bleeding, fistula, and stricture formation. Bleeding was defined as melena, hematemesis, or coffee ground vomitus. Perforation and/or fistula formation was diagnosed using chest radiography, computed tomography or endoscopy. Stricture was indicated by symptoms of dysphagia, regurgitation, or odynophagia with confirmation via endoscopy or upper gastrointestinal radiography. Systemic complications included aspiration injury, respiratory failure, hepatic injury, renal injury, sepsis, and disseminated

intravascular coagulation. Hepatic injury was defined as serum alanine aminotransferase or aspartate aminotransferase levels elevated to three times the normal upper limit. Renal injury was defined as serum creatinine level >1.4 mg/dL without other noted renal diseases.

## Cytokine measurements using multiplex immunoassay

Apart from blood samples of 22 patients with caustic gastrointestinal injury, blood samples from 18 healthy controls were included for comparison. Cytokine measurement was performed using the Bio-Plex Human cytokine assay kit (Bio-Rad Laboratories, Hercules, CA), namely IL-2, IL-5, IL-8, IL-9, IL-12, IL-13, interferon-gamma inducible protein-10, macrophage inflammatory protein-1 beta, regulated upon activation, normal T cell expressed and presumably secreted and tumor necrosis factor-alpha, according to the manufacturer's instructions. Samples from 18 healthy controls were included for comparison. In brief, 50 ul antibody-coupled beads per well were added to the flat bottom plate and wash two times. Then, 50 ul plasma sample was incubated with antibody-coupled beads for 30 minutes at room temperature. After washing three times to remove unbound materials, the beads were incubated with 25 ul biotinylated detection antibodies for 30 minutes at room temperature. After washing away the unbound biotinylated antibodies for three times washes, the beads were incubated with 50 ul streptavidin phycoerythrin for 10 minutes at room temperature. Following removal of excess streptavidin phycoerythrin for three times washes, the beads were resuspended in 125 ul assay buffer. Beads were read on the Bio-Plex suspension array system, and the data were analyzed using Bio-Plex Manager software version 6.0.

## Statistical analysis

Demographic data of continuous variables were expressed as mean ± standard deviation (range), and those of categorical variables were presented as numbers with percentages. All statistical tests were two-sided and were performed using IBM Statistical Product and Service Solutions (SPSS), version 22 (IBM, Armonk, New York, USA). We conducted Mann-Whitney tests for comparing continuous variables. The categorical variables were assessed via Pearson $\chi^2$ tests, Fisher's exact tests, or Fisher-Freeman-Halton test. The Kaplan-Meier method was used for the survival analysis, with the difference between survival curves assessed via the log-rank test. A P value < 0.05 was considered to be statistically significant.

## Acknowledgments

Thank you to all the colleagues from the Department of Gastroenterology and Hepatology, Department of Psychiatry, Department of Emergency Medicine, Department of Medical Image and Intervention, Division of Trauma and Emergent Surgery, and Division of General Surgery of Linkou Chang Gung Memorial Hospital for helping us care for the patients.

## Author Contributions

**Conceptualization:** Hao-Tsai Cheng, Chen-June Seak, Tzung-Hai Yen.

**Data curation:** Hao-Tsai Cheng, Chen-June Seak, Chien-Cheng Cheng, Tsung-Hsing Chen, Chang-Mu Sung, Shih-Ching Kang, Yu-Jhou Chen, Chip-Jin Ng, Chao-Wei Lee, Shu-Wei Huang, Hsin-Chih Huang, Tzung-Hai Yen.

**Formal analysis:** Hao-Tsai Cheng, Chien-Cheng Cheng, Tsung-Hsing Chen, Chang-Mu Sung, Shih-Ching Kang, Yu-Jhou Chen, Chip-Jin Ng, Chao-Wei Lee, Shu-Wei Huang, Hsin-Chih Huang, Tzung-Hai Yen.

**Funding acquisition:** Hao-Tsai Cheng, Tzung-Hai Yen.

**Investigation:** Hao-Tsai Cheng, Chen-June Seak, Chien-Cheng Cheng, Tsung-Hsing Chen, Chang-Mu Sung, Shih-Ching Kang, Yu-Jhou Chen, Chip-Jin Ng, Chao-Wei Lee, Shu-Wei Huang, Hsin-Chih Huang, Tzung-Hai Yen.

**Methodology:** Hao-Tsai Cheng, Chen-June Seak, Chien-Cheng Cheng, Tsung-Hsing Chen, Chang-Mu Sung, Shih-Ching Kang, Yu-Jhou Chen, Chip-Jin Ng, Chao-Wei Lee, Shu-Wei Huang, Hsin-Chih Huang, Tzung-Hai Yen.

**Project administration:** Hao-Tsai Cheng, Chen-June Seak, Tzung-Hai Yen.

**Resources:** Hao-Tsai Cheng.

**Software:** Yu-Jhou Chen.

**Supervision:** Tzung-Hai Yen.

**Writing – original draft:** Hao-Tsai Cheng.

**Writing – review & editing:** Chen-June Seak, Chien-Cheng Cheng, Tsung-Hsing Chen, Chang-Mu Sung, Shih-Ching Kang, Yu-Jhou Chen, Chip-Jin Ng, Chao-Wei Lee, Shu-Wei Huang, Hsin-Chih Huang, Tzung-Hai Yen.

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
