## [Decision Letter · Decision Letter 0]

7 Jul 2021

PONE-D-21-18809

Profiling of inflammatory cytokines in patients with caustic gastrointestinal tract injury

PLOS ONE

Dear Dr. Yen,

Thank you for submitting your manuscript to PLOS ONE. After careful consideration, we feel that it has merit but does not fully meet PLOS ONE’s publication criteria as it currently stands. Therefore, we invite you to submit a revised version of the manuscript that addresses the points raised during the review process.

We look forward to receiving your revised manuscript.

Kind regards,

Hsu-Heng Yen

Academic Editor

PLOS ONE

Journal Requirements:

2. Please describe in your methods section how capacity to consent was determined for the participants in this study. Please also state whether your ethics committee or IRB approved this consent procedure. If you did not assess capacity please briefly outline why this was not necessary in this case.

Reviewers' comments:

Reviewer's Responses to Questions

**Comments to the Author**

1. Is the manuscript technically sound, and do the data support the conclusions?

Reviewer #1: Partly

Reviewer #2: Partly

2. Has the statistical analysis been performed appropriately and rigorously? 

Reviewer #1: I Don't Know

Reviewer #2: Yes

3. Have the authors made all data underlying the findings in their manuscript fully available?

Reviewer #1: Yes

Reviewer #2: Yes

4. Is the manuscript presented in an intelligible fashion and written in standard English?

Reviewer #1: Yes

Reviewer #2: Yes

5. Review Comments to the Author

Reviewer #1: Authors have made a good attempt to find the role of interleukins in caustic injury of GI tract. I have following comments to make:

1. Small sample size. How the study subjects were selected since both the groups (mild and severe) had equal number of patients.

2. Study was done in 2018 why it was not published in last 3 years.

3. Abbreviation of interleukin (IL) can be used instead of repeating it several times.

4. What was the aim/objective of the study.?

5. Kindly mention the aim in abstract as well.

6. Modify table 1. It is confusing as the authors have used "/"and several numbers are written in a single column. This will confuse the readers.

7. Authors should mention whether the study is cross-sectional or follow up study. They have vaguely mentioned in discussion and conclusion that it was a short follow up but the details on how the follow up was done and what was the outcome in each patients. They have mentioned that 2 patients died in 3-6 months which is very vague sentence and questions the credibility of the follow up.

8. Mention some more details regarding the type of surgery /indication/ timing etc in these patients.

9. What is the implication of this study on the treatment of patients as it is mentioned that steroids have no role in the treatment. How will the patients be benefitted by measuring the IL levels.

10. Mention the novelity of study and its role in future research.

Reviewer #2: This study analysis cytokine response after caustic gastrointestinal injury, and though the change in cytokine level might be the biomarker for outcome prediction. It showed elevation of such as interleukin (IL) 2,5, 8, 9, 12 interferon gamma, and tumor necrosis factor(TNF) alpha after caustic injury compaired with healthy controls. Especially IL-2 and TNF-alpha were higher in severe caustic injury group. But there is no further analysis for cytokine change in complication such as systemic complication, or GI complication such as stricture. It is hard to conclude that there is no benefit of steroid for prevent esophageal stricture owing to no significance difference in IL-10 in caustic injruy group. One study had analysis Infliximab, TNF alpha antibody, in experimental alkali burns of the oesophagus in the rat. Knowing the cytokine response related complication after caustic injury may evoke more study for treatment.

6. PLOS authors have the option to publish the peer review history of their article (what does this mean?). If published, this will include your full peer review and any attached files.

Reviewer #1: **Yes: **Shivaraj Afzalpurkar

Reviewer #2: No

---

## [Author Response · Author response to Decision Letter 0]

1 Oct 2021

Reviewer #1: Authors have made a good attempt to find the role of interleukins in caustic injury of GI tract. I have following comments to make:

Point 1. Small sample size. How the study subjects were selected since both the groups (mild and severe) had equal number of patients.

Response: Thank you for the comment. All patients with caustic gastrointestinal tract injury seen at our hospital between March 2018 and October 2018 were included in the study. Patients who refused signed informed consent were excluded from the study. The equal number of patients of both groups was by coincidence.

Point 2. Study was done in 2018 why it was not published in last 3 years.

Response: Thank you for the comment. We apologize for the delay in manuscript preparation.

Point 3. Abbreviation of interleukin (IL) can be used instead of repeating it several times.

Response: Thank you for the comment. The term has been abbreviated.

Point 4. What was the aim/objective of the study?

Response: Thank you for the comment. This study aimed to investigate the cytokine profiling of patients with caustic substance ingestion, and analyzed the differences between patients with severe and mild caustic gastrointestinal tract injury.

Point 5. Kindly mention the aim in abstract as well.

Response: Thank you for the comment. The aim of this study has been included in the Abstract.

Point 6. Modify table 1. It is confusing as the authors have used "/"and several numbers are written in a single column. This will confuse the readers.

Response: Thank you for the comment. Table 1 has been revised.

Point 7. Authors should mention whether the study is cross-sectional or follow up study. They have vaguely mentioned in discussion and conclusion that it was a short follow up but the details on how the follow up was done and what was the outcome in each patients. They have mentioned that 2 patients died in 3-6 months which is very vague sentence and questions the credibility of the follow up.

Response: Thank you for the comment. We confirmed that this is a cross-sectional study. We apologize for the vague sentence, the inappropriate sentence has ben revised.

Point 8. Mention some more details regarding the type of surgery /indication/ timing etc in these patients.

Response: Thank you for the comment. A new Table 5 has been created to describe the clinical courses of the 5 patients who underwent surgery. The indications for operations were mainly esophageal stricture or gastric outlet obstruction.

Point 9. What is the implication of this study on the treatment of patients as it is mentioned that steroids have no role in the treatment. How will the patients be benefitted by measuring the IL levels?

Response: Thank you for the comment. None of our patients were treated with glucocorticoids. Beneficial effects of steroid on preventing stricture formation in severe caustic ingestion, is controversial according to published literatures. Nevertheless, the findings of this translational study provide clinical significance. Our analysis found that caustic gastrointestinal tract injury is characterized by a systemic inflammatory response that involves elevated levels of circulating inflammatory cytokines and neutrophils. The analysis showed increased circulating levels of inflammatory cytokines (IL-2, IL-5, IL-8, IL-9, IL-12, IL-13, interferon-gamma inducible protein-10, macrophage inflammatory protein-1 beta, regulated upon activation, normal T cell expressed and presumably secreted and tumor necrosis factor-alpha) in patients with caustic gastrointestinal injury than healthy controls. Furthermore, the neutrophil counts and circulating levels of certain cytokines (IL-2 and tumor necrosis factor-alpha) were higher in patients with severe than mild caustic gastrointestinal injury. Additionally, since circulating levels of IL-2 and tumor necrosis factor-alpha were higher in nonsurvivors than in survivors, these two cytokines might have clinical potential as promising prognostic markers for caustic gastrointestinal injury. In this context, clinical determination of the circulating inflammatory response, particularly IL-2 and tumor necrosis factor-alpha levels, could serve as a valuable adjunct to physiological predictors for the prediction of poor outcome.

Point 10. Mention the novelty of study and its role in future research.

Response: Thank you for the comment. This is supposed to be the first study to profile inflammatory cytokines in patients with caustic gastrointestinal tract injury. Caustic injury of the upper gastrointestinal tract causes severe mucosal damages and leads to excessive levels of inflammatory cytokines and neutrophils in the peripheral blood. The findings of this translational study provide clinical significance. Our analysis found that caustic gastrointestinal tract injury is characterized by a systemic inflammatory response that involves elevated levels of circulating inflammatory cytokines and neutrophils. The analysis showed increased circulating levels of inflammatory cytokines (IL-2, IL-5, IL-8, IL-9, IL-12, IL-13, interferon-gamma inducible protein-10, macrophage inflammatory protein-1 beta, regulated upon activation, normal T cell expressed and presumably secreted and tumor necrosis factor-alpha) in patients with caustic gastrointestinal injury than healthy controls. Furthermore, the neutrophil counts and circulating levels of certain cytokines (IL-2 and tumor necrosis factor-alpha) were higher in patients with severe than mild caustic gastrointestinal injury. Additionally, since neutrophil counts and circulating levels of IL-2 and tumor necrosis factor-alpha were higher in nonsurvivors than in survivors, these two cytokines might have clinical potential as promising prognostic markers for caustic gastrointestinal injury. In this context, clinical determination of the circulating inflammatory response, particularly IL-2 and tumor necrosis factor-alpha levels, could serve as a valuable adjunct to physiological predictors for the prediction of poor outcome. 

Reviewer #2: This study analysis cytokine response after caustic gastrointestinal injury, and though the change in cytokine level might be the biomarker for outcome prediction. It showed elevation of such as interleukin (IL) 2,5, 8, 9, 12 interferon gamma, and tumor necrosis factor (TNF) alpha after caustic injury compaired with healthy controls. Especially IL-2 and TNF-alpha were higher in severe caustic injury group. But there is no further analysis for cytokine change in complication such as systemic complication, or GI complication such as stricture. It is hard to conclude that there is no benefit of steroid for prevent esophageal stricture owing to no significance difference in IL-10 in caustic injruy group. One study had analysis Infliximab, TNF alpha antibody, in experimental alkali burns of the oesophagus in the rat. Knowing the cytokine response related complication after caustic injury may evoke more study for treatment.

Response: Thank you for the comment. We are agreed that this study is limited by small sample size and lacking serial cytokine measurement. This important limitation has been included to the Discussion section.

The most obvious limitation in this study was small sample size. Moreover, there was no further analysis for cytokine change in complication such as systemic complication, or gastrointestinal complication such as stricture. Therefore, it was difficult to conclude that there is no benefit of steroid for prevent esophageal stricture owing to no significance difference in IL-10 between two groups of caustic patients. Notably, anti-tumor necrosis factor alpha treatment with infliximab did not ameliorate the degree of fibrosis in alkali burns of the esophagus in the rat. Knowing the cytokine response related complication after caustic injury may evoke more study for treatment. Further studies are necessary to elucidate the influences of immune responses on the clinical outcomes of corrosive patients.

---

## [Decision Letter · Decision Letter 1]

2 Nov 2021

Profiling of inflammatory cytokines in patients with caustic gastrointestinal tract injury

PONE-D-21-18809R1

Dear Dr. Yen,

We’re pleased to inform you that your manuscript has been judged scientifically suitable for publication and will be formally accepted for publication once it meets all outstanding technical requirements.

Kind regards,

Hsu-Heng Yen

Academic Editor

PLOS ONE

Additional Editor Comments (optional):

  This article was reviewed by two expert reviewers and it can be accepted  in its present form.

Reviewers' comments:

Reviewer's Responses to Questions

**Comments to the Author**

1. If the authors have adequately addressed your comments raised in a previous round of review and you feel that this manuscript is now acceptable for publication, you may indicate that here to bypass the “Comments to the Author” section, enter your conflict of interest statement in the “Confidential to Editor” section, and submit your "Accept" recommendation.

Reviewer #1: All comments have been addressed

2. Is the manuscript technically sound, and do the data support the conclusions?

Reviewer #1: Yes

3. Has the statistical analysis been performed appropriately and rigorously? 

Reviewer #1: I Don't Know

4. Have the authors made all data underlying the findings in their manuscript fully available?

Reviewer #1: Yes

5. Is the manuscript presented in an intelligible fashion and written in standard English?

Reviewer #1: Yes

6. Review Comments to the Author

Reviewer #1: All the comments are addressed by the authors. As authors mentioned, clinical

determination of the circulating inflammatory response, particularly IL-2 and tumor

necrosis factor-alpha levels, could serve as a valuable adjunct to physiological

predictors for the prediction of poor outcome.

7. PLOS authors have the option to publish the peer review history of their article (what does this mean?). If published, this will include your full peer review and any attached files.

Reviewer #1: **Yes: **Shivaraj Afzalpurkar

---

## [Editor Report · Acceptance letter]

8 Nov 2021

PONE-D-21-18809R1 

Profiling of inflammatory cytokines in patients with caustic gastrointestinal tract injury 

Dear Dr. Yen:

I'm pleased to inform you that your manuscript has been deemed suitable for publication in PLOS ONE. Congratulations! Your manuscript is now with our production department. 

Kind regards, 

on behalf of

Dr. Hsu-Heng Yen 

Academic Editor

PLOS ONE